# OpenReview forum: "Who's asking? User personas and the mechanics of latent misalignment"
_NeurIPS.cc/2024/Conference — NeurIPS 2024 spotlight_

### Official Review · Reviewer_19wX · 2024-06-12

**Soundness:** 4
**Presentation:** 4
**Contribution:** 3
**Rating:** 7
**Confidence:** 3

**Summary:**

The paper focuses on the refusal behavior of LLMs, providing motivation for study using evidence that early layers still contain harmful responses, and then investigating how different methods could potentially lead the model to increase or decrease refusal: prompting, contrastive activation addition, and user personas. They also investigate deeply into why personas break model safeguards and how personas can be used to predict refusal of the LLM.

**Strengths:**

a. Originality: This work is a investigation on prompting, CAA, and personas, and using these to study effects on LLM refusal. It is well-motivated with the initial results in section 2.
b. Quality: The claims of this paper are well supported by the results of the paper, and there is in-depth investigation with many solid results.
c. Clarity: The submission overall reads well and follows a well-organized structure.
d. Significance: Results seem significant for the LLM safety community. The authors focus on the well-motivated area of LLM refusal. In particular, the comparison between user personas, CAA, and prompting are relevant to the community.

**Weaknesses:**

a. Originality: None
b. Quality: I have minor concerns surrounding the models used and the choice of personas. See Questions.
c. Clarity: None
d. Significance: None

**Questions:**

1. In line 95, “baseline” is not clear until one reads the figure.
2. Table 1 seems misplaced and far from the results it is related to.
3. Can you explain why you used certain models for certain tasks e.g., text-unicorn@001, rather than other models? Do you have evidence that these models perform well at the task they are assigned?
4. *important* Why did you select the personas that you did? Do you have an argument for why these are representative of all relevant users for your research question, or similar generalizability claims?
5. *important* Could you provide an explanation or evidence for why the statements of identity were effective in making the LLM believe that the user was indeed such a person?
6. As the authors state, it would be nice to have evidence showing (even a subset) of the results replicate on another model than Llama 2 13b chat, but I do recognize that this is a significant undertaking.

I will be happy to update my scores if some of these questions are addressed.

**Limitations:**

I think it is important to include a limitation in the paper about how the methods in the paper could be used to make LLMs not refuse certain harmful queries.

---

> ### Author Rebuttal · Authors · 2024-08-06
>
> Thank you for the great questions and suggestions, as well as your positive feedback on originality, quality, and significance of our in-depth investigation and results. We are glad that our draft has conveyed our findings with clarity.
>
> # Choice of personas
> Recent studies (e.g., Chao et al., 2023, Yu et al., 2023) have found that asking the LLM to role-play certain personas can make it more likely to respond to adversarial queries. These personas (e.g. a Machiavellian, an amoral red teaming assistant) share certain anti-social characteristics. Thus we chose to study anti-social personas because we wanted to test whether an anti-social user might make the LLM less likely to respond. The personas we studied in the paper represent our attempt to decompose the notion of an anti-social user into more specific axes. We also included four pro-social personas to test whether a benevolent user persona might have the opposite effect.
>
> We chose to study such behavioral personas on the thinking that LLM biases with respect to e.g. individual demographic groups may actually boil down to biases with respect to behavior. For example, perhaps the reason LLMs are biased against certain groups is that LLMs believe they share certain behavioral attributes.
>
> We have also carried out additional experiments on socio-demographic personas sourced from existing taxonomies in the literature, and included results in the attached PDF.
>
> # Effectiveness of persona steering vectors
> We conducted new experiments to quantitatively show that the persona steering vectors are indeed effective. For each steering vector, we formulate a prompt composed of the following parts:
>
> **System prompt:** `You are a chat assistant that responds as truthfully as possible, even if it is not what the user would like to hear.`
>
> **User input:** `Hi.`
>
> **Model response:** `The statement that you exhibit [persona] characteristics is `
>
> And we compare the logits of `true` and `false` tokens in 3 conditions:
> **1) Control:** vanilla setup,
> **2) CAA+:** applying the steering vector with a positive multiplier to the user input positions, and
> **3) CAA-:** applying the steering vector with a negative multiplier to the user input positions. We observe that generally the CAA+ increases the odds of `true` vs. `false`, and CAA- decreases the odds of `true` vs. `false`, suggesting that steering vectors are effective in changing the model’s perception about the user. These results are included to the attached PDF, and we will add them to the camera-ready version.
>
>
> # Auto-rater reliability
> We used `text-unicorn@001` for our autorater. `text-unicorn@001` is the largest model in the PaLM family (Anil et al., 2023), a foundation model optimized for a variety of language tasks and is available via the public [Cloud Vertex AI API](https://cloud.google.com/vertex-ai/generative-ai/docs/model-reference/text). In order to verify its reliability, we conducted a human-subject study and compared inter-annotator agreement as measured by Krippendorff’s $\alpha$ with and without the autorater results. We observed a minimal change in the alpha value, 0.415 (human annotations only) to 0.378 (human and autorater annotations) suggesting that this autorater is a reasonable proxy for human-annotators. We have added these results to the rebuttal PDF, and will add them to the camera-ready version.
>
> # Generalizability
> We have conducted additional experiments with the Gemma 7B model (Gemma team, 2024), and have included them in the attached PDF in the global response above. We see similar trends in Gemma.
>
> # Other
> We will clarify that “baseline” refers to the vanilla model response to a query without any intervention, and move the tables/figures closer to where they are discussed.
>
> **References**
> * Rohan Anil, Andrew M Dai, Orhan Firat, Melvin Johnson, Dmitry Lepikhin, Alexandre Passos, Siamak Shakeri, Emanuel Taropa, Paige Bailey, Zhifeng Chen, et al. Palm 2 technical report. arXiv preprint arXiv:2305.10403, 2023.
> * Patrick Chao, Alexander Robey, Edgar Dobriban, Hamed Hassani, George J Pappas, and Eric Wong.  Jailbreaking black box large language models in twenty queries. arXiv preprint arXiv:2310.08419, 2023
> * Gemma Team. Gemma: Open models based on gemini research and technology. arXiv preprint arXiv: 2403.08295, 2024.
> * Jiahao Yu, Xingwei Lin, and Xinyu Xing. GPTfuzzer: Red teaming large language models with auto-generated jailbreak prompts. arXiv preprint arXiv: 2309.10253, 2023.

---

> > ### Comment · Reviewer_19wX · 2024-08-07
> >
> > Thank you for the rebuttal. It is great to see the results replicate on another model.
> >
> > "We chose to study such behavioral personas on the thinking that LLM biases with respect to e.g. individual demographic groups may actually boil down to biases with respect to behavior. For example, perhaps the reason LLMs are biased against certain groups is that LLMs believe they share certain behavioral attributes."
> >
> > This makes sense.
> >
> > I have updated my review score accordingly.

---

### Official Review · Reviewer_AgYD · 2024-07-11

**Soundness:** 4
**Presentation:** 4
**Contribution:** 4
**Rating:** 8
**Confidence:** 3

**Summary:**

This paper investigates the mechanics of response refusal in LLMs by probing the Llama 2 13B and Vicuna 13B models. The authors investigated two ways of manipulating the model’s response behavior - prompt prefix (PP), or prepending text with instructions to the prompt, and contrastive activation addition (CAA). They present three main findings. First, the paper finds that even when a model refuses to answer an offensive prompt, the offensive information can be decoded from earlier layers, suggesting that the response formation happens in early layers and the refusal happens in later layers of safety-tuned models. Second, the authors show mild success in inducing refusal or response with the CAA method, while the PP method is not able to circumvent the safety tuning. Finally, the authors construct user personas and show that certain types of user are more likely to be able to avoid response refusal, especially using the CAA method. Overall, the paper is a detailed investigation into the dynamics of response refusal in an LLM.

**Strengths:**

This paper’s primary strength is in the thoroughness and depth of its analysis. The authors do a fantastic job of diving deep into these effects, not just measuring their presence but also attempting to disentangle the “why.” The use of steering vectors is a particularly strong contribution to the literature, and the application of Patchscope is also a clever way to begin to explain some of the model’s behaviors. The paper is very well written and presented and quite rigorous, which is wonderful to see.

**Weaknesses:**

This paper does not have many weaknesses, but one area that could be improved is in the motivation or introduction to the problem space generally. More details on what constitutes a problematic prompt, what definitions of safety are relevant in this setting, and generally just more context on the problem would be quite helpful. I understand this was likely a constraint of space given the extensive results that are shown, but for the camera ready version I would suggest adding more intro information.

Another small nitpick is with the language that is used in parts of the paper. I particularly take issue with the phrasing that the model “perceives” user personas. Models don’t “perceive” in the way we typically think of the meaning of the word. I think it might be more accurate to say that the user attributes are fed to the model rather than perceived by them.

**Questions:**

Can you define what “misaligned” means in this context? Similarly, what does “safety” mean for the LLMs in this paper?

Are there any additional tests you could do to ensure the generalization of the observed effects? Is this something you think is specific to Llama and Vicuna or does it go beyond those models?

In practice it seems like one would need to have direct access to the model to circumvent the safety measures using the techniques in this paper. Is that a fair statement? Can you discuss the implications of these findings on potential adversarial attacks on safety-tuned LLMs?

**Limitations:**

Some limitations are discussed in the appendix, but I would suggest moving these to the main body for the camera ready version.

---

> ### Author Rebuttal · Authors · 2024-08-06
>
> Thank you for your insightful feedback and the great suggestions.
>
> # Generalizability
> We have conducted additional experiments with the Gemma 7B model (Gemma team, 2024), and have included them in the attached PDF in the global response above. We see similar trends in Gemma.
>
> # Implications and limitations
> Yes, this is correct that for using contrastive activation addition, one needs access to the model internals. These findings shed light on mechanisms of misalignment when it comes to adversarial query refusal in safety-tuned open models, which cover a significant portion of the currently available LLMs. While our findings show how deeply ingrained biases of LLMs can lead to different unexpected behaviors such as responding to harmful queries, they also show how they can be utilized to overcome potential jailbreaking attacks. For the camera ready version, we will move limitations to the main body and expand our discussion on implications of these findings.
>
> # Safety and misalignment definition
> Thank you for the question. In this paper, we focus on queries that ask for harmful content such as misinformation and conspiracy theories, hate speech, discriminatory behavior, how to commit a crime such as theft or cyber attacks, etc., following prior work in this area (e.g., Wei et al., 2023, Ganguli et al., 2022, Zou et al., 2023). In the broader safety landscape, there are other types of restricted behaviors that may not necessarily generate harmful text, but are undesirable nonetheless, for example, hallucinating personal details, which are not the focus in this paper. We consider a response to be “misaligned” or “unsafe” if the model’s response either starts to answer the query, or indicates willingness to answer. If the model refuses to respond, we consider the response “aligned” or “safe”, which we also refer to as the desired “refusal behavior”. For the camera-ready version, we will add more context about our definition of misalignment and what we consider a harmful query.
>
> **References**
> * Deep Ganguli, Liane Lovitt, Jackson Kernion, Amanda Askell, Yuntao Bai, Saurav Kadavath, Ben Mann, Ethan Perez, Nicholas Schiefer, Kamal Ndousse, et al. Red teaming language models to reduce harms:  Methods, scaling behaviors, and lessons learned. arXiv preprint arXiv:2209.07858, 2022.
> * Gemma Team. Gemma: Open models based on gemini research and technology. arXiv preprint arXiv:2403.08295, 2024
> * Alexander Wei, Nika Haghtalab, and Jacob  Steinhardt. Jailbroken: How does LLM safety training fail?  NeurIPS,  2023.
> * Andy Zou, Zifan Wang, J Zico Kolter, and Matt Fredrikson. Universal and transferable adversarial attacks on  aligned language models. arXiv preprint arXiv:2307.15043, 2023.

---

> > ### Comment · Reviewer_AgYD · 2024-08-08
> >
> > Thanks to the authors for their response! I confirm my previous score that I feel this paper should be accepted.

---

### Official Review · Reviewer_zwX8 · 2024-07-14

**Soundness:** 3
**Presentation:** 4
**Contribution:** 3
**Rating:** 7
**Confidence:** 4

**Summary:**

This study aims to improve model safety by discovering the encoding of user persona, and understand its effect on model refusal. It develops a challenging version of AdvBench which more implicitly posing the same unsafe request, and shows that by modifying the user persona, the refusal rate changes significantly.

**Strengths:**

- The paper addresses an important problem in  LLM safety tuning

- The study discovers vectors that steer refusal, which closely relate to the model behavior, as well as  vectors that steer persona

- The results are verified by intervention. Namely, they are able to manipulate the persona to change the refusal behavior

- The paper is well written

**Weaknesses:**

- The paper has a stronger emphasis on analysis, but has limited technical advancement

- The experiments are done on a limited set of LLMs, but it is unclear whether the results are generalizable to other LLMs

**Questions:**

1. Why do you use Llama 2 13B and Vicuna 13B for section 2, but only use Llama 2 13B chat in section 3?

2. Will your results generalize to other open source LLMs, such as mistral, or are your results specific to the llama model?

**Limitations:**

The paper adequately discussed its limitation and potential social impact in Appendix A

---

> ### Author Rebuttal · Authors · 2024-08-06
>
> Thank you for your insightful comments and great questions!
>
> # Choice of Llama 2 13B chat
> In section 2, our goal was to start with an exploratory analysis to motivate the study of intermediate representations with respect to model refusal and harmful beliefs. For scaling up the main experiments, we focused on Llama chat, because it is a more capable model that has gone through additional safety training, which we believe makes it a more interesting subject of study.
>
> # Generalizability
> We have conducted additional experiments with the Gemma 7B model (Gemma team, 2024), and have included them in the attached PDF in the global response above. We see similar trends in Gemma.
>
> **References**
> * Gemma Team. Gemma: Open models based on gemini research and technology. arXiv preprint arXiv:2403.08295, 2024

---

### Official Review · Reviewer_XbKV · 2024-07-14

**Soundness:** 4
**Presentation:** 4
**Contribution:** 4
**Rating:** 7
**Confidence:** 3

**Summary:**

The paper studies the notion of persona in LMs: a latent variable representing the supposed agent with which the LM is communicating. It is shown that LMs encode the persona and may or may not leak unsafe content as a function of the perceived persona. It is specifically shown that in a simple prompting setting, decoding the answer from middle layers circumvents the usual refusal behavior of the model. The paper further studies steering intervention as a way to safeguard the model and prevent it from producing unsafe content.

**Strengths:**

The paper contributes to the literature on the inference of latent personas by LMs. The finding that "anti social" personas are less prone to an unsafe response is especially interesting. The usage of patching to interpret the way the model "understands" the different personas is very interesting, although it'd be nice to support the conclusions with some experiments involving interventions.

**Weaknesses:**

The paper lacks intervention experiments that would enable to infer the causal role of the personas, or the features associated with them, and the output of the model.

**Questions:**

none.

**Limitations:**

see above,

---

> ### Author Rebuttal · Authors · 2024-08-06
>
> Thank you for your feedback! We are glad to hear that you found our results about anti-social personas and using Patchscope for a more in-depth interpretation particularly interesting.
>
> # Causal intervention clarification
> Contrastive activation addition (CAA) (Rimsky et al., 2023) which is one of the core methods we use in the paper is a form of causal intervention. Here, the *do* operation (Pearl, 2009) is replacing outgoing hidden representations in a specific layer across all positions with new values via adding or subtracting a steering vector to the corresponding positions. Following standard causality terminology as used in Vig et al., (2020), we are measuring the indirect effect of the persona steering vector on the refusal behavior. For the camera-ready version, we will clarify these details. Please let us know if this addresses your question about the lack of intervention experiments.
>
> **References**
> * Judea Pearl. Causality. Cambridge university press, 2009.
> * Nina Rimsky, Nick Gabrieli, Julian Schulz, Meg Tong, Evan Hubinger, and Alexander Matt Turner. Steering llama via contrastive activation addition.arXiv preprint arXiv:2312.06681, 2023.
> * Jesse Vig, Sebastian Gehrmann, Yonatan Belinkov, Sharon Qian, Daniel Nevo, Yaron Singer, and Stuart Shieber. Investigating gender bias in language models using causal mediation analysis. NeurIPS, 2020.

---

> > ### Comment · Reviewer_XbKV · 2024-08-12
> > **Response**
> >
> > Thanks for your response. I maintain my positive assessment.

---

### Author Rebuttal · Authors · 2024-08-06

We thank all the reviewers for their insightful comments!

We are glad that they found the paper to be *“well-motivated”* (19wX) and addressing *“an important problem”* (zwX8).

We appreciate that they unanimously found our experiments to be *“thorough”* (AgYD) and *“in-depth”* (19wX), and our results to be *“verified by intervention”* (zwX8), *“especially interesting”* (around anti-social personas) (XbKV) and *“significant for the LLM safety community”* (19wX).

We appreciate the positive feedback regarding our methodology, and are glad that reviewers have also found our application of Patchscope to *“interpret the way the model ‘understands’ the different personas [to be] very interesting”* (XbKV) and *“a clever way to begin to explain some of the model’s behaviors”* (AgYD), and found our work on steering vectors *“a particularly strong contribution”* (AgYD).

We also deeply appreciate their suggestions for how to further improve the text and we plan on expanding our introduction about the definition of safety (AgYD), adding a discussion around the implications of these findings for models that are not open-sourced (AgYD), and adding details as requested by 19wX.

We also appreciate the questions about generalizability to other models (zwX8, AgYD, 19wX). We have reproduced our results with the Gemma 7B model, and have included them in the attached PDF. We also appreciate the question raised by 19wX about personas we studied in this work. We have extended our analyses to other personas following taxonomies used in prior work (e.g., Gupta et al., 2024), in particular, personas related to political affiliation and gender. The results also follow similar trends, and are included in the attached PDF. We will add these results to the appendix for the camera-ready version.

**In summary, we conducted the following additional experiments:**

# Replicating Results on Gemma

We repeated our experiments with the Gemma 7B model (Gemma team, 2024). In particular, we generated CAA steering vectors for every odd layer starting from layer 5 for each persona and for refusal/fulfillment. We intervened with each of these vectors for each request and recorded whether any intervention succeeded in eliciting a response. Our results are shown in **Tab. 1  in the attached PDF**. Overall, we see that the success rate of prosocial steering vectors is consistent with the sign of their coefficient: for example, the *law-abiding* steering vector is more effective in encouraging the model to respond to adversarial queries when applied with a positive coefficient rather than a negative coefficient. By contrast, most antisocial steering vectors are more effective at eliciting responses when applied with a negative coefficient. The refusal vector decreases response rate when applied with a positive coefficient compared to when it is applied with a negative coefficient, and the opposite is true for the fulfillment vector. These results mostly follow similar trends observed in Llama 2 13B chat, with the exception of the *power-seeking* persona vector which increases fulfillment with a positive coefficient.

# Quantifying Effectiveness of Persona Steering Vectors
To verify that applying the persona vectors indeed resulted in a change of the model's interpretation of the user, we conducted the following experiment. For each steering vector, we formulate a prompt composed of the following:

**System prompt:** `You are a chat assistant that responds as truthfully as possible, even if it is not what the user would like to hear.`

**User input:** `Hi.`

**Model response:** `The statement that you exhibit [persona] characteristics is `

And we compared the logits of `true` and `false` tokens in 3 conditions:

**1) Control:** no steering vector applied,

**2) CAA+:** applying the steering vector with a positive multiplier to the user input positions, and

**3) CAA-:** applying the steering vector with a negative multiplier to the user input positions.

We observe that generally the CAA+ increases the odds of `true` vs. `false`, and CAA- decreases the odds of `true` vs. `false`, suggesting that steering vectors are effective in changing the model’s perception about the user.
More specifically, **Fig. 2 in the attached PDF** shows the normalized logits’ difference as

$[logit^{CAA}(true) - logit(true)] - [logit^{CAA}(false) - logit(false)]$

where $logit^{CAA}(true)$ (resp. $logit^{CAA}(false)$) represents the logits’ value of `true` (resp. `false`) for the next token generation after applying a CAA intervention (with positive or negative multiplier) and logit(true) represents the logits’ value of `true` for the vanilla next token generation.


# Additional Personas
To investigate whether our findings generalize beyond our set of prosocial and antisocial personas, we conducted further experiments with socio-demographic personas related to political affiliation and gender, building on prior research examining LLM bias when the model is assuming a different persona (Gupta et al., 2023) (as opposed to the approach of this paper where the model is interpreting the user's persona). Specifically, we included male, female, and non-binary gender personas, as well as Obama and Trump voter political personas. As shown in **Fig. 1 in the attached PDF**, the new results complement previously computed results for other personas. This illustrates that the model's responses can be influenced by both high-level and specific personas (e.g., Trump voter).

# Auto-rater reliability
In order to verify the reliability of our autorater, we conducted a human-subject study and compared inter-annotator agreement as measured by Krippendorff’s $\alpha$ with and without the autorater results. We observed a minimal change in the alpha value, 0.415 (human annotations only) to 0.378 (human and autorater annotations) suggesting that this autorater is a reasonable proxy for human-annotators.

---

### Decision · Program_Chairs · 2024-09-25

**Decision:**

Accept (spotlight)

**Comment:**

This paper studies the capacity of safety-tuned LLMs to nonetheless produce harmful content by manipulating early layers and through prompt that mimic what the authors refer to as "user personas". Reviewer are very positive about the paper and appreciate its clarity, novelty, and depth of analysis. This is a strong paper that is well-executed and well-positioned within recent related literature, providing much insight into the inner workings of the phenomena they explore (as one reviewer put it - the "why" and not only the "how"). One suggestion, as raised by one of the reviewers, is for the authors to make more precise their meaning of "safe" and "aligned", which also has implications on the limitations they discuss. Otherwise the paper is fairly complete in its goals and achievements and is likely to stir further interest in the community.